# Low-Cost Smartphone Photogrammetry Accurately Digitises Positive Socket and Limb Casts

Sean Cullen [1,*], Ruth Mackay [1], Amir Mohagheghi [2] and Xinli Du [1]

1 Department of Mechanical and Aerospace Engineering, College of Engineering Design and Physical Sciences, Brunel University, Kingston Lane, Uxbridge UB8 3PH, UK; ruth.mackay@brunel.ac.uk (R.M.); xinli.du@brunel.ac.uk (X.D.)

2 Sport, Health & Exercise Sciences, College of Health, Medicine and Life Sciences, Brunel University, Kingston Lane, Uxbridge UB8 3PH, UK; amir.mohagheghi@brunel.ac.uk

* Correspondence: sean.cullen3@brunel.ac.uk

**Abstract:** Digitising prosthetic sockets and moulds is critical for advanced fabrication techniques enabling reduced lead times, advanced computer modelling, and personalised design history. Current 3D scanners are expensive (>GBP 5000) and difficult to use, restricting their use by prosthetists. In this paper, we explore the use and accuracy of smartphone photogrammetry (<GBP 1000) as an accessible means of digitising rectified socket moulds. A reversed digital twin method was used for evaluating accuracy, in addition to simplified genetic algorithms to identify an optimal technique. The identified method achieved an accuracy of 99.65% and 99.13% for surface area and volume, respectively, with an interclass coefficient of 0.81. The method presented is simple, requiring less than ten minutes to capture using twenty-six photos. However, image processing time can take hours, depending on the software used. This method falls within clinical limits for accuracy, requires minimal training, and is non-destructive; thus, it can be integrated into existing workflows. This technique could bridge the gap between digital and physical workflows, helping to revolutionise the prosthetics fitting process and supporting the inclusion of additive manufactured sockets.

**Keywords:** prosthetics; sockets; scanning; photogrammetry; low cost; digital twin; genetic algorithms





## 1. Introduction

Conventionally, lower limb prosthetic sockets are fabricated using hand casting and rectification techniques. Despite the fact that digital design and fabrication techniques and their benefits have been explored in the literature over a relatively long period of time, traditional techniques still dominate [1–7]. This is likely due to the inaccessibility of the technology, difficulty regarding the skills gap, and challenges related to its integration with existing workflows. Shape and surface topography are critical for user comfort, where inaccuracies in scanning can result in increased pressure and pain in the produced sockets [8–10]. Many devices, such as MRI scanners, laser scanners, and optical scanners, have been used to scan limbs and sockets [11–15]. These scanners are costly and often require specialised training, hindering their use in limb centres.

D. Solav. et al. introduced a low cost method for residuum scanning using an array of digital cameras. This method was further improved upon by M. Barreto. et al., who achieved a scanning error of <1.93 mm using a 3D printed residuum mould [16,17]. While this method reflected low costs, it required a specialised cylindrical camera array which introduced barriers to its inception in clinical practice. By contrast, smartphone photogrammetry offers a method for digitisation without the requirement for specialised equipment. Smartphone photogrammetry was first introduced by A. Hernandez and E. Lemaire for scanning socket interiors [18]. However, similar work by the present authors noted inaccuracies in scanning internal volumes near the base of the socket wall when using photogrammetry because of the limited number of angles at which the photographs

were taken [19]. The inaccuracies noted would potentially lead to reproduced sockets with inferior fit and reduced comfort for amputees. For positive socket moulds however, there are no restrictions from the imaging perspective.

A recent survey indicated that 70% of prosthetists experience challenges in their socket production work flows, with casting and rectification being reported as problematic by 50% and 56% of clinicians, respectively [19]. As these fields are still dominated by hand techniques, digital integration in these areas could have the greatest impact. However, the artisan skill perfected by clinicians through years of practice with hand rectification is not directly transferable to the digital space. This could explain the fact that when surveyed by the American Board of Certification for Orthotists and Prosthetist in 2014, only 23% of prostheses were reportedly fabricated using computer aided design (CAD) and computer aided manufacturing (CAM) techniques [20]. The scanning of rectified casts, as explored in this paper, may be better suited to the current industry environment that is direct limb scanning. Because scanning is a non-destructive process, a scan of a rectified mould, acting as a backup, can be taken prior to its destruction during the socket lamination process. Alternatively, a socket could be directly produced from the scan data, using either centralised manufacturing facilities or 3D printing [5,11,21,22].

Despite the authors' prior work showing that photogrammetry was not a suitable scanning method for socket interiors, the restrictions on photographing position are not present when scanning positive residuum casts or socket moulds [23]. However, to fairly evaluate photogrammetry as a positive cast scanning technique, every possible arrangement of photos should be considered. This is not practical, as the computational and time costs would be prohibitive. Instead, a series of experiments can be conducted to evaluate select photogrammetry experiments. Therefore, the authors aimed to use simple genetic algorithms (SGA) to determine a series of experiments to identify an optimal photogrammetry technique. In this context, an optimal photogrammetry technique is both accurate and simple to conduct. SGA was used to significantly reduce the number of scan experiments required to identify an optimal technique.

This project is the first instance in the literature in which photogrammetry was used to scan positive prosthetic casts. The methods of digital file comparison and technique optimisation using SGA have been used previously, but not for positive prosthetic casts. The relative accessibility of photogrammetry and the simplicity of the technique could help bridge the skills gap hindering current digital techniques.

## 2. Materials and Methods

### 2.1. Physical Model Preperation

A male rectified socket mould was obtained from an existing socket using plaster of Paris (PoP). The socket was destroyed in this process to preserve the mould. The cast was covered in an unbranded printed nylon sleeve, placed on a pedestal, and photographed employing random camera positions using an iPhone 12 (Apple, Cupertino, CA, USA) on automatic settings. The nylon sleeve had a thickness of $0.17 \pm 0.01$ mm across the fabric, with a brim thickness of $0.40 \pm 0.03$ mm. The photos were processed using Autodesk ReCap (Autodesk, San Francisco, CA, USA). The 3D mesh file was scaled using a point-to-point scaling tool and the marks on the metal ruler. The file was trimmed to remove unnecessary mesh and cut to the base of the cast using visual alignment and the slice-and-fill tool. The processed mesh was then used as the digital twin model control, which was 3D printed using a Creality CR-10 (Creality, Shenzhen, China) with a nozzle size of 0.8 mm and a 0.36 mm layer height. The printed model acted as the physical component of the digital twin (physical control model). Some material was trimmed from the bottom of the model to reduce the print time; however, the socket profile was left intact. This process is highlighted in Figure 1.

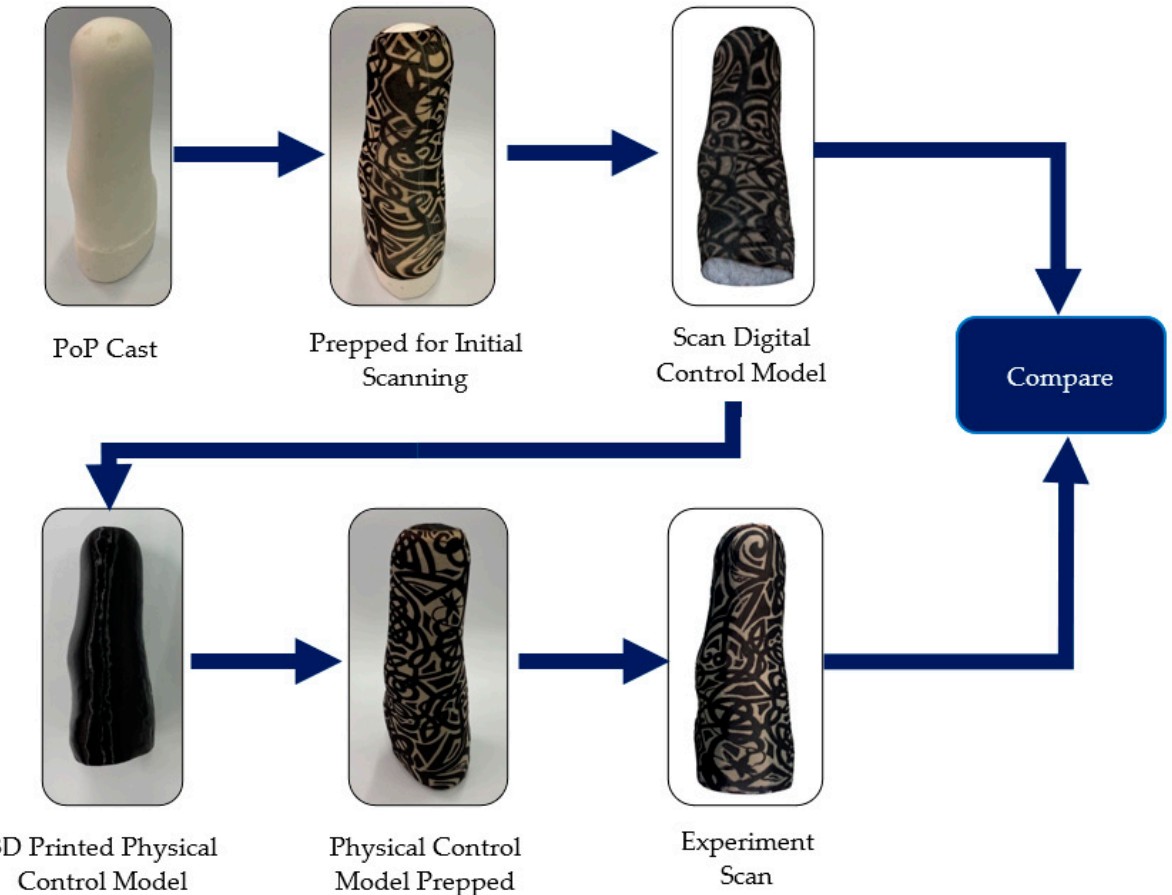

**Figure 1.** Experimental process stages, including the creation of control models. The prepped stages show the patterned nylon sleeve, and the scan files show the digital visualisation of the resulting 3D scan.

### 2.2. Optimisation Study

The 3D printed control model (physical twin) was mounted onto a pedestal and covered in a patterned nylon sleeve, as in the previous scan. The sleeve exhibited a different pattern, but used the same colours. A 30 cm and a 15 cm metal ruler were placed next to the model for scaling. A printed protractor was placed under the model to indicate 15° increments. A total of 288 photos were captured of the 3D model, using the points designated by Table 1, at 15° increments ($\theta$) based on a cylindrical coordinate system. Camera positions were controlled using a tripod for vertical height and ruler for radial distance. The angular position of the camera was aligned visually with the marks on the protractor at the model base. A visual representation of the experimental setup and the location of the photos is shown in Figure 2.

**Table 1.** Position of photos taken for each gene set.

| Gene Photo Set | Horizontal Distance r (cm) | Height from Base of Model h (cm) |
|:---:|:---:|:---:|
| 1 | 150 | −10 |
| 2 | 150 | 20 |
| 3 | 150 | 50 |
| 4 | 100 | −10 |
| 5 | 100 | 20 |
| 6 | 100 | 50 |
| 7 | 50 | −10 |
| 8 | 50 | 20 |
| 9 | 50 | 50 |

**Table 1.** *Cont.*

| Gene Photo Set | Horizontal Distance r (cm) | Height from Base of Model h (cm) |
|:---:|:---:|:---:|
| 10 | 30 | 20 |
| 11 | 30 | 50 |
| 12 | 10 | 50 |

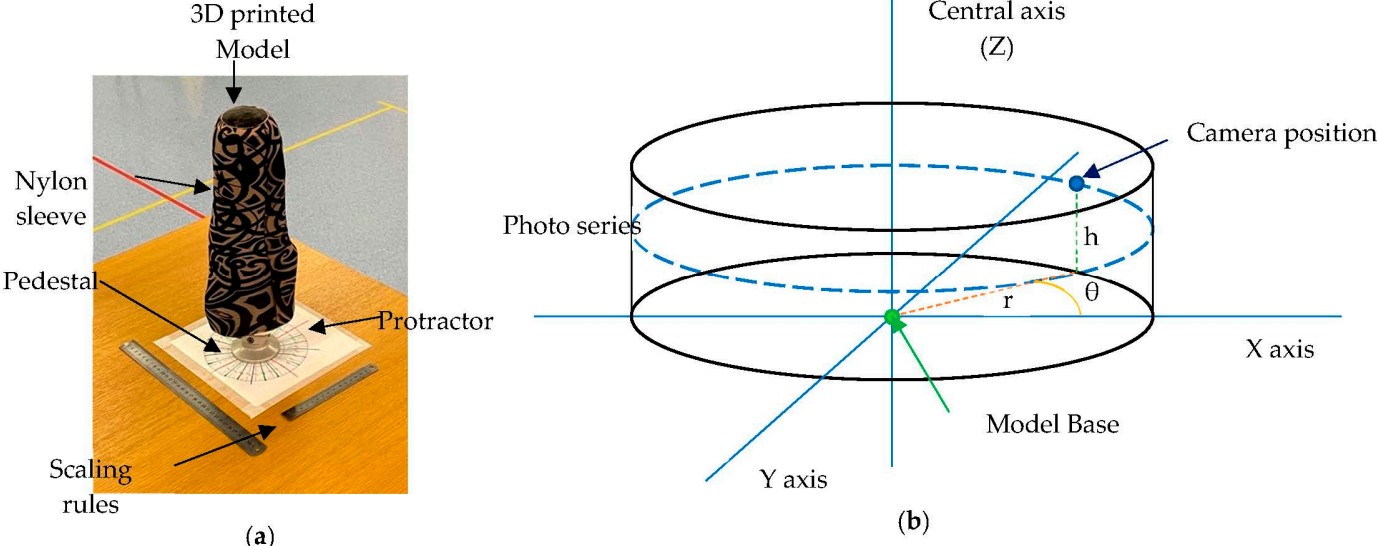

(**a**)        (**b**)

**Figure 2.** (**a**) Experimental photography setup and (**b**) cylindrical coordinate system used to define camera position, measured in height (h), radial distance (r), and angle (θ).

For accuracy and simplicity, simplified genetic algorithms (SGAs) were used to identify an optimal scanning technique. Due to the large number of possible combinations of photos and the limitations of ReCap, which can only process 100 photos, SGAs were chosen for selecting the groups of photos which should be collated for the scanning experiments. SGAs allow for testing a wide range of random photo sets, as well targeted optimisation through incremental improvements, without requiring every possible combination be tested.

Fitness Equation (1), which equally rewarded the generational ranking of average scan accuracy and the number of photos in the gene set, was employed. The fitness score is calculated using the scanning accuracy regarding the percentage of volume and surface area, as well as the number of photos. Experiments with higher fitness exhibit higher accuracy and or a lower number of photos required. Lower photo counts indicated a simpler photography technique.

Due to the time required to run experiments (up to two hours), the study was limited to five generations. This was in line with the methods used in prior work [23]. After the selection process, an additional photo reduction was carried out on the most fit gene set of the 5th generation. This was achieved by the reduction of two angle increments for each active gene in the set. The identified optimal gene set was then retested using the offset starting angle to test the constancy in the scan accuracy.

$$\text{Fitness} = \frac{\text{Rank}\left(\frac{\text{Volume} + \text{SurfaceArea}}{2}\right) + \text{Rank}(\text{Number of Photos})}{2} \tag{1}$$

The resulting scans from each experiment were scaled using the longest visible length scale on a 30 cm ruler placed within the photographed volume, allowing for alternate measurement points, if sections of the ruler were blurred or distorted. A secondary ruler was also included to check the model sizing after scaling. The meshes were cropped to the baseline of the model. Each scan model was compared to the digital control model in regards to volume and surface area to evaluate the accuracy as a difference in percentage

from that of the control model. The fitness for each experiment was calculated using Equation (1), providing a ranking score within each generation.

Each experimental gene set consisted of twelve gene pairs made up of two gene bits. This gene code controlled which photo sets would be included in the scan (G columns) and the angle increment (A columns) that would be used (Table 2). Due to the simplicity of the genes, MS Excel (Microsoft, Redmond, WA, USA) was used to create, store, and compare their fitness. Initially, the genes were generated at random for the first generation, which consisted of fifteen gene sets. Subsequent generations consisted of the top two fittest genes from the previous generation (24LCC1-13, 24LCC1-7); two genes generated from breeding the fittest two genes with a crossover point after the sixth gene pair (24LCC2-1 to 2); three mutated genes from each of the prior winners (24LCC2-3 to 8); and five additional randomly generated gene sets (24LCC2-9 to 13). A combination of gene breeding and mutation was used to optimise the genes within local optima, while the introduction of random genes allowed for continual exploration of the design space for the global optima.

**Table 2.** The 2nd generation of the gene sets, with the active genes in each set shown in grey.

| Gene Set Name | A1 | G1 | A2 | G2 | A3 | G3 | A4 | G4 | A5 | G5 | A6 | G6 | A7 | G7 | A8 | G8 | A9 | G9 | A10 | G10 | A11 | G11 | A12 | G12 | Photo Count |
|---|---|---|---|---|---|---|---|---|---|---|---|---|---|---|---|---|---|---|---|---|---|---|---|---|---|
| 24LCC1-13 | 45 | 1 | 30 | 0 | 45 | 0 | 60 | 0 | 60 | 1 | 45 | 0 | 45 | 0 | 30 | 0 | 45 | 1 | 45 | 1 | 15 | 0 | 15 | 0 | 30 |
| 24LCC1-7 | 60 | 0 | 60 | 0 | 15 | 0 | 60 | 0 | 60 | 1 | 45 | 0 | 45 | 0 | 15 | 1 | 60 | 0 | 15 | 0 | 15 | 1 | 60 | 1 | 60 |
| 24LCC2-1 | 45 | 1 | 30 | 0 | 45 | 0 | 60 | 0 | 60 | 1 | 45 | 0 | 45 | 0 | 15 | 1 | 60 | 0 | 15 | 0 | 15 | 1 | 60 | 1 | 68 |
| 24LCC2-2 | 60 | 0 | 60 | 0 | 15 | 0 | 60 | 0 | 60 | 1 | 45 | 0 | 45 | 0 | 30 | 0 | 45 | 1 | 45 | 1 | 15 | 0 | 15 | 0 | 22 |
| 24LCC2-3 | 45 | 1 | 30 | 0 | 45 | 0 | 60 | 1 | 60 | 1 | 45 | 0 | 45 | 0 | 30 | 0 | 45 | 1 | 45 | 1 | 15 | 0 | 15 | 0 | 36 |
| 24LCC2-4 | 30 | 1 | 30 | 0 | 45 | 0 | 60 | 0 | 60 | 1 | 45 | 0 | 45 | 0 | 30 | 0 | 45 | 1 | 45 | 1 | 15 | 0 | 30 | 0 | 34 |
| 24LCC2-5 | 45 | 1 | 30 | 0 | 45 | 0 | 60 | 1 | 60 | 1 | 45 | 0 | 45 | 0 | 30 | 0 | 60 | 1 | 45 | 1 | 15 | 1 | 15 | 0 | 58 |
| 24LCC2-6 | 60 | 0 | 60 | 0 | 15 | 0 | 60 | 0 | 60 | 1 | 45 | 0 | 45 | 0 | 15 | 1 | 45 | 0 | 15 | 0 | 15 | 1 | 45 | 1 | 62 |
| 24LCC2-7 | 60 | 1 | 60 | 0 | 15 | 0 | 60 | 0 | 60 | 1 | 45 | 0 | 45 | 1 | 15 | 1 | 60 | 0 | 15 | 0 | 15 | 1 | 60 | 1 | 74 |
| 24LCC2-8 | 60 | 0 | 60 | 0 | 30 | 0 | 60 | 0 | 60 | 1 | 45 | 0 | 45 | 1 | 15 | 1 | 60 | 1 | 30 | 0 | 15 | 1 | 60 | 1 | 74 |
| 24LCC2-9 | 45 | 1 | 45 | 1 | 15 | 0 | 60 | 0 | 45 | 1 | 60 | 1 | 30 | 0 | 30 | 0 | 15 | 1 | 30 | 0 | 60 | 0 | 60 | 0 | 54 |
| 24LCC2-10 | 45 | 0 | 15 | 0 | 45 | 1 | 60 | 0 | 15 | 0 | 45 | 0 | 30 | 1 | 15 | 0 | 30 | 0 | 15 | 1 | 60 | 0 | 45 | 0 | 44 |
| 24LCC2-11 | 45 | 1 | 60 | 0 | 15 | 0 | 45 | 0 | 60 | 1 | 30 | 0 | 45 | 1 | 15 | 1 | 30 | 0 | 45 | 0 | 30 | 1 | 60 | 1 | 64 |
| 24LCC2-12 | 30 | 1 | 60 | 1 | 15 | 0 | 60 | 1 | 30 | 0 | 30 | 1 | 60 | 0 | 60 | 0 | 30 | 0 | 45 | 0 | 60 | 0 | 15 | 0 | 36 |
| 24LCC2-13 | 15 | 0 | 30 | 0 | 30 | 0 | 15 | 0 | 30 | 0 | 45 | 0 | 30 | 0 | 60 | 1 | 30 | 1 | 15 | 0 | 30 | 1 | 30 | 1 | 42 |

For breeding, the gene crossover point was chosen as the midpoint between photo sets (6th gene pair): in Table 2, this is shown by the gene sets 24LCC2-1 and 2, when compared with their respective parents, 24LCC1-13 and 7. For mutations (24LCC2-3 to 8), any gene bit from the parent gene set had a 10% chance of changing value: for column G, this mutation would change a value of 1 to 0 or vice versa, including or removing a photo series from the final gene set. In contrast, gene bits in the angle columns (A), if mutated, showed an even chance of increasing or decreasing by one increment (15°, 30°, 45°, 60°), except for 15° and 60° as the upper and lower increments, which could only change to 30° and 45°, respectively, as is noted for gene pair 12 on 24LCC2-6 when compared to its parent LCC1-7. Additionally, if mutations occurred only on the inactive gene pairs, then there would be no change to the overall experiment, and these mutations were re-generated. An example of this is shown in 24LCC2-8, where the A3 gene bit was set at 30° compared to 15° of its parent (24LCC1-7), but this gene set was still included because it remained unique, exhibiting additional mutations on the 8th and 9th gene pairs. Using trial and error, the mutation rate was set at 10% to encourage mutations on one or two gene pairs, without distorting the overall gene makeup.

The continual introduction of random gene sets for each generation helped to avoid early convergence into a local optimum, ensuring an overall better inter-generational optimisation. When generated, each new gene set was compared to the existing gene sets to avoid duplicates. This comparison was conducted only on active genes because the gene sets could be different on the inactive genes, making the experiments identical. A visual basic (VBA) code was written to automatically extract the relevant photos required by each gene set to be meshed using ReCap, simplifying the upload process.

A second model was generated from a sperate PoP mould, and a subsequent physical twin was 3D printed in green thermal polyurethane (TPU), referred to as Model 2. A research assistant with no experience in photogrammetry photographed both physical twins using the same method to identify inter-operator repeatability. The research assistant was provided with the list of photos required, the relative camera position, a tape measure for measuring distances, and instructed not to alter objects in the imaging environment.

### 2.3. Scan Model Analysis

Once identified, the optimal technique scan file was imported to Fusion 360 (Autodesk, CA, USA) and aligned with the control model. Both models were then segmented, based on parallel planes to the control model base, into 10 mm increments. The segment volumes, top surface areas, and loop length are visualised in Figure 3.

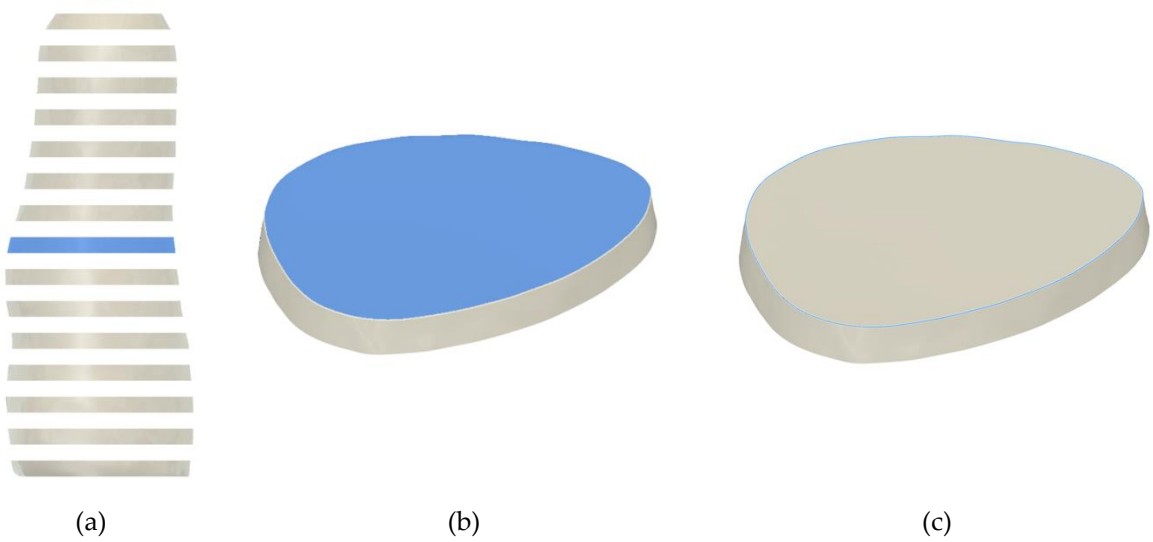

(a)            (b)            (c)

**Figure 3.** Visualisation of comparative metrics: (**a**) the scan model, with alternate segments hidden and a single segment highlighted in blue; (**b**) a single segment, with the top surface highlighted; (**c**) a loop length of the same segment.

One of the metrics used in the literature to obtain clinically acceptable socket fit deviations between socket copies is the mean radial error (MRE). A fabricated socket with an MRE > 0.25 mm can be deemed clinically unacceptable [24]. Using the loop length, the mean radius for each section was calculated, assuming the loop length to be the circumference of a perfect circle. The MRE was then calculated by comparing the mean radius for each section between the control and scan model.

### 3. Results

Of the seventy-four unique experiments conducted, fifteen scans were rejected due to incomplete or prominent extrusions from the model mesh. Across the experiments conducted, the average volume and surface area accuracy compared to those of the control model were 97.74% and 99.02%, respectively, with an average photo count of fifty-two. The final selected gene set achieved an accuracy of 98.09% (volume) and 99.56% (surface area), requiring twenty-six photos from three radial position sets. The cylindrical positions of the final gene set referred to as 24LCC6-3 were 100 cm (r) and 20 cm (h), at 45° increments; 50 cm (r) and 50 cm (h), at 60° increments; and 30 cm (r) and 50 cm (h), at 30° increments. The optimised reduction in the number of photos required is shown in Figure 4, and fluctuations in the leader of generation three (experiment 26) show that the transition from accuracy is prioritised by the fitness function in relation to the reduced photo count.

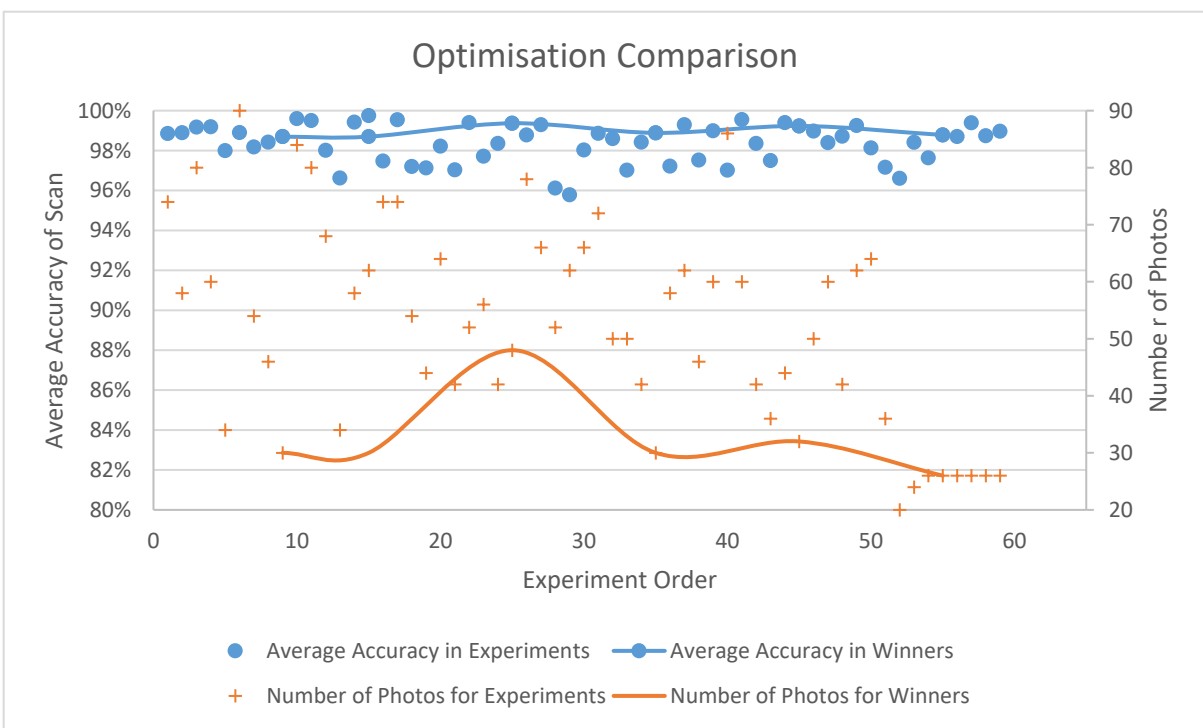

**Figure 4.** Overview of experiments conducted, in which accuracy is measured by the averaged accuracy of the scan volume and the surface area.

The repeatability of the scanning technique was calculated to be within 0.2% for volume, SA, and height by changing the offset for the radial position of the images by 15°, 30°, and 60°. In addition, using the technique, two different models were photographed and scanned, by both an experienced and inexperienced person, resulting in an interclass correlation coefficient (ICC) of 0.81, indicating good inter-operator reliability [14].

In Figure 5, the segment volumes, top surface areas, and loop length are compared between the scan and control model. Across the central model region, between 20–280 mm, the average percentage difference between the scan and the control model was 0.75% for volume, 0.82% for surface area, 0.40% for loop length, with a 0.2 mm MRE.

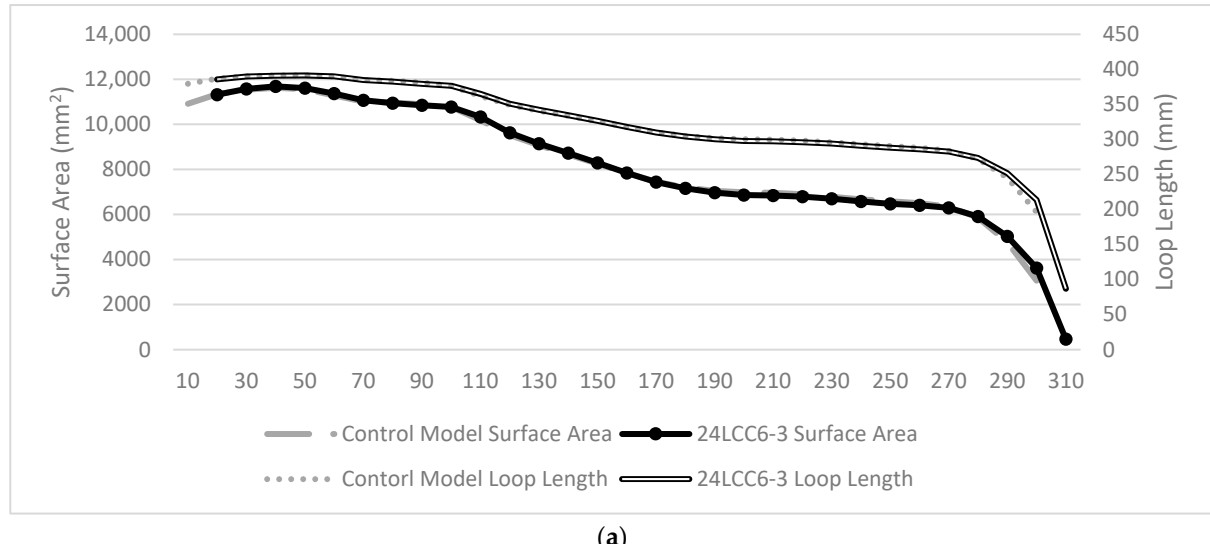

(**a**)

**Figure 5.** *Cont.*

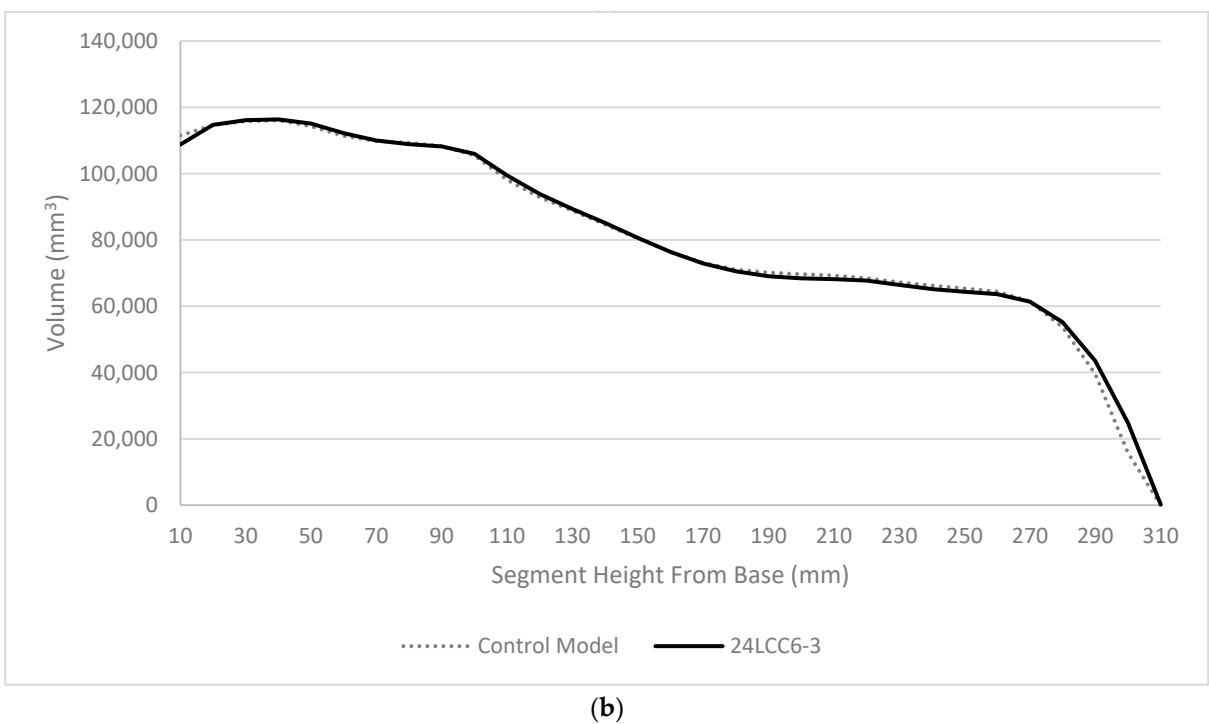

**(b)**

**Figure 5.** Comparison of scan segments to control model for (**a**) surface area and loop length and (**b**) volume.

## 4. Discussion

The aim of this experiment was to identify an optimal photogrammetry scanning method for positive socket and residuum casts which could be easily incorporated into current practice. The genetic algorithms used were able to identify an image set consisting of only 26 photos from three radial positions, which achieved an accuracy for volume and surface area of 98.09% and 99.56%, respectively, increasing to 99.13% and 99.65% during a repeat test. This is a significant reduction in the number of photos required when compared to the average of 52 across all experiments. The most accurate experiment set tested achieved an accuracy of 99.59% and 99.81% for volume and surface area, respectively; however, it required 62 photos from four of the twelve possible radial positions. The fitness equation used in this experiment assigned equal weighting to the number of photos and the averaged accuracy for scan volume and surface area. The time required to capture a full scan was more significantly impacted by the number of radial positions needed compared to the number of angular increments required. A fitness equation with different weightings may have achieved a slightly more accurate scan without significantly impacting the photography time. The specific time required to acquire the images was, however, beyond the scope of this study.

The optimised method presented in this paper was intended to render inclusion in clinical practice as simple as possible. As such, the use of smartphones, simple measuring and scaling devices, and a printed nylon sleeve result in both low equipment costs and setup times. The equipment employed was generic, but the use of specialised sleeves and tools would be of benefit for increasing the accuracy around the distal cavity and reducing the photography time. Cast preparation before photographs were captured took approximately thirty seconds, and the preparation time included the sleeve application. The time required to capture photos of the device was approximately five minutes, which is comparable to that required by other 3D scanning techniques [14]. However, the processing time was significantly longer. For this study, Autodesk ReCap Photo was used, and the meshes were processed using cloud computing. The scans required up to two hours to mesh following the uploading of the photos, but this was heavily dependent on server workload, with the shortest scan taking 5–10 min. Other software that processes images on

a local machine could be used; however, this would limit the process to completing one scan at a time, in contrast to the abilities of cloud-based systems to process multiple scans simultaneously. It is also worth noting that ReCap is a paid software service, which would present a cost to clinics.

While the optimisation process led to overall improvements in the accuracy and simplicity of the photogrammetry technique, the average scan accuracy was ~97%. Therefore, it was difficult for the fitness equation to reward increases in accuracy, as the percentage difference between experiments was minimal. This explains why there is a fluctuation in the optimisation scores after generation three, as the reduction in photos produces larger differences than do the changes in accuracy. Unfortunately, due to the manual input requirements of the photogrammetry software, it was not possible to test additional generations, nor to test the impact of different fitness equations or SGA parameters.

Compared to the photogrammetry methods explored previously for sockets, scanning the residuum model achieved a reasonably high reliability [23]. Out of the initial seventy-four experimental scans, ten were rejected due to significant surface anomalies or other model deformities. However, when the final selected experiment was rerun at varying start angles to check for reliability, all of the scans returned similar results (Table 2). It should also be noted that capturing additional photos to those recommended for the technique did not reduce the scan quality, nor did it provide a significant increase in accuracy.

When testing for repeatability between different users, both an experienced and inexperienced photographer achieved an ICC of 0.81, accuracies of >99% in height and surface area, and an accuracy of >95% in volume using the identified scanning method. This suggests that the method was not heavily reliant upon an individual's skill, provided that appropriate basic verbal instructions are given. However, this was limited to two operators scanning two models; thus, further repeatability testing would provide more insight into the clinical utility of this technique.

The potential sources of error in this experiment can be divided into two categories: digital and physical. The digital errors included the scaling factor, the mesh slice point and angle, and the digital alignment of the models. The physical errors consisted of the thickness added by the nylon sleeve and the resolution of the 3D printed model. For this study, the physical errors were considered negligible. It should be noted that in some scans, a ridge caused by the nylon sleeve stitching could be identified at the distal end of the residuum. While this area was located in the distal cavity and may not affect clinical application, the ridge could be removed by the use of a nylon sock with a closed end. At the time of this study, the authors could not locate a closed sock suitable in regards to either pattern or thickness; however, directly marking the cast should provide the same results. Further research could identify whether the normal identifying marks added by clinicians during the casting process would be sufficient for scanning, which would further increase the clinical compatibility of the technique.

In addition to the errors regarding physical accuracy, the impact of the digital scaling factor was estimated at ~0.3%. The mesh slice angle and the position for removing the surrounding area scanned was not directly measurable and would not impact the clinical effectiveness of the scan. Across the sixty-four experiments accepted, the average deviation in the sliced model height was 1.9 mm, or 0.63%; however, this deviation was a compound of the slice angle and height and the scaling factor.

The identified photogrammetry technique could integrate well with traditional clinician workflows, without significant interruption. Initially, this could enable the capturing of digital rectification history and ultimately aid in the transition to a completely digital workflow. Currently, the digital manufacturing stages are not compatible with hand techniques, requiring the re-casting of any 3D-printed cast backup scans. It should also be noted that the work by J. Sanders et al. queried the accuracy and quality of prosthetics produced from centralised fabrication facilities [25]. This could be avoided with the use of direct 3D printed of sockets, but limited research has been conducted on the strength of these sockets [26,27].

## 5. Conclusions

Seventy-four unique experimental sets were analysed to determine the optimal methodology for the smartphone photogrammetry of a rectified residuum cast. The preparation time for the model using a patterned nylon sleeve was approximately thirty seconds, with approximately five minutes required to capture the photographs, but software solving added a significant amount of time. The method employed in this study proved to be reliable, achieving a volume accuracy of 98.09%, which is within the clinically acceptable limit of 5% [28]. The ICC was 0.81 across two models and two operators. This photogrammetry method was conducted on a rectified residuum cast, but it is equally applicable to a positive amputee cast. Overall, the proposed method is simple, suitable for limb digitisation in a clinical setting, and is available at a relatively low cost compared to specialised alternatives. Future work is recommended to investigate whether prostheses created using photogrammetry scans are as comfortable as the hand laminated versions.

**Author Contributions:** Conceptualization, S.C.; methodology, S.C.; software, S.C.; validation, S.C.; formal analysis, S.C.; investigation, S.C.; resources, S.C.; data curation, S.C.; writing—original draft preparation, S.C.; writing—review and editing, S.C., A.M., R.M. and X.D.; visualization, S.C.; supervision, S.C. and X.D.; project administration, S.C.; funding acquisition, S.C. All authors have read and agreed to the published version of the manuscript.

**Funding:** This work was supported by EPSRC through a studentship for Sean Cullen (EP/R512990/1).

**Institutional Review Board Statement:** Ethical review and approval were waived for this study by the Brunel Ethics team. The casts used were taken from a decommissioned socket belonging to one of the authors.

**Informed Consent Statement:** Not applicable.

**Data Availability Statement:** The data presented in this study are available on request from the corresponding author. The data are not publicly available due to privacy.

**Conflicts of Interest:** The authors declare no conflict of interest.

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
