# Peer review of "Low-Cost Smartphone Photogrammetry Accurately Digitises Positive Socket and Limb Casts"

_prosthesis, doi:10.3390/prosthesis5040095_

Round 1

Reviewer 1 Report

Comments and Suggestions for Authors

Thanks for the opportunity to review this interesting and well written paper looking at optimizing the way a smart phone can be used to digitize a residual limb mould.

I have a few comments below that I hope the authors will find useful to add additional clarity to the study.

Some general comments are below followed by some detailed comments by line number.

General:

Firstly, in the title they refer to socket moulds, but they digitize residual limb plaster moulds in the study – consider revising?

Secondly, there is a lot of technical writing about ‘optimization’ in this study, but this is a clinical journal, it may be worth the authors stating in plain language in the introduction what optimal means in this context? As short as possible, minimising computing power? Accuracy? Use context specific language if possible.

Thirdly, maybe just me not knowing too much about these algorithms (sorry if that is the case!), but a large section of the methods is devoted to the algorithms used, it would be good if the authors could provide a brief few sentences on why it was necessary to use the algorithm and what they were using it for in plain language- I think the aim was to use the algorithm to identify which combinations of phot positions were best (most accurate, minimized number of photos etc?) and used the fitness equation to score each combination?

Finally, in the data analysis, can the authors add a section in the methods to explain how the measure of ‘accuracy’ in % terms that is reported throughout the paper was calculated? In addition, the ICC calculation performed is only on 2 casts so is likely underpowered. This should be acknowledged.

Detailed:

Intro:

Line 32 ‘Many devices, such as MRI scanners…’ – This sentence doesn’t make sense is it incomplete?

Line 50 – This is also an area that has been highlighted by clinicians as challenging and where they would be open to technology to help in their workflows. Here is one suggested article along those lines : https://pubmed.ncbi.nlm.nih.gov/36112475/

Line 60: ‘ It should be noted that work by J Sanders…’ – these last 2 sentences could maybe move to the discussion, they distract from a well written paragraph above. Limitations will be more relevant in discussion.

Methods:

Line 82: Can the authors add a bit more detail about the use of ReCap to make a 3D mesh, there isn’t enough detail here to allow reproduction of methods.

Line 99: Was the same sleeve used on every mould?

Line 100: Can the authors clarify, were the photos taken by hand just using the protractor and rulers as guides or was there some sort of rig to position things?

Line 115: ‘SGA was chosen to design…’do authors means design locations for photos to be taken or something else? Incremental improvements in what? Try to use plain language here.

121: ? Typo ‘in in percentage…’

121: Can the authors clarify what makes a ‘good’ fitness from your equation? Are you trying to Minmise errors and number of photos?

126: typo missing ‘of’ in ‘reduction of two’

127: What does constancy of results mean? No change in accuracy?

136: When authors talk about evaluating accuracy and fitness can they clarify – you compared the volume and surface area of the control model to the volume and surface area of the photographed model – how did you compare fitness?

177: Not clear why the second model was generated? Why do this? Where is it used in the results?

Results:

There are several degree symbols appearing as ‘o’ throughout para 1 of results.

You have 2 figure 4s, need to update one to fig 5 and update in text references to fit.

Discussion:

Where authors talk about using Autodesk ReCap it may also be worth mentioning that this is payed software that will also need to be purchased by the clinic?

Line 295: ‘taking additional photos to the recommended sets did not reduce the scan quality…’ – can the authors clarify if it improved accuracy?

Line 310: Authors talk about not finding a better sock – what about just marking the cast directly? Is this possible?

I wish the authors all the best with the writing up and continued work in this important area

Comments on the Quality of English Language

Nil

Reviewer 2 Report

Comments and Suggestions for Authors

The paper entitled “Low Cost Smartphone Photogrammetry Accurately Digitises Socket Moulds" investigates the use and accuracy of smartphone photogrammetry as an accessible means of digitising rectified socket mould. From my point of view, the topic is of great interest. But some minor comments:

·         To enhance the abstract, it would be beneficial to include the potential implications of these findings on the application of additively manufactured materials.

·         I would have liked the author to add analysis on laser scanning to reconstruction (https://doi.org/10.3390/sym15091776; https://doi.org/ 10.1007/s40430-023-04231-9)

·         It is not very conventional and appropriate to end the introduction with a reference to an article.

·         It is better to end your introduction by talking about the novelty of the work presented.

·         Regarding experimental setup: The scales are not visible in the image in Figure 2, if you have another photo where they are visible it would be interesting.

·         Please do not end subsections (e.g. 2.2) with a Figure. Subsections should end with a summary text.

·         The conclusions must discuss the limitations of their study.

·         And suggest future research directions

Comments on the Quality of English Language Text review, inappropriate sentences and general English review.  

Round 2

Reviewer 2 Report

Comments and Suggestions for Authors

Improvements introduced after the first version. accept in current state